# CRAYM: Neural Field Optimization
# via Camera RAY Matching

**Liqiang Lin**[1]    **Wenpeng Wu**[1]    **Chi-Wing Fu**[2]    **Hao Zhang**[3]    **Hui Huang**[1*]

[1]College of Computer Science and Software Engineering, Shenzhen University
[2]Department of Computer Science and Engineering, The Chinese University of Hong Kong
[3]School of Computing Science, Simon Fraser University
linliqiang2020@gmail.com  wenpengggg@gmail.com
cwfu@cse.cuhk.edu.hk  haoz@sfu.ca  hhzhiyan@gmail.com

## Abstract

We introduce *camera ray matching* (CRAYM) into the joint optimization of camera poses and neural fields from multi-view images. The optimized field, referred to as a feature volume, can be "probed" by the camera rays for novel view synthesis (NVS) and 3D geometry reconstruction. One key reason for matching camera rays, instead of pixels as in prior works, is that the camera rays can be parameterized by the feature volume to carry both geometric and photometric information. Multi-view consistencies involving the camera rays and scene rendering can be naturally integrated into the joint optimization and network training, to impose physically meaningful constraints to improve the final quality of both the geometric reconstruction and photorealistic rendering. We formulate our per-ray optimization and *matched ray coherence* by focusing on camera rays passing through *keypoints* in the input images to elevate both the efficiency and accuracy of scene correspondences. Accumulated ray features along the feature volume provide a means to discount the coherence constraint amid erroneous ray matching. We demonstrate the effectiveness of CRAYM for both NVS and geometry reconstruction, over dense- or sparse-view settings, with qualitative and quantitative comparisons to state-of-the-art alternatives.

## 1  Introduction

Recent advances on multi-view 3D reconstruction have been propelled by the emergence of neural fields [39], including implicit functions [42, 36, 24, 33] and radiance fields (NeRF) [26, 44, 13, 1, 10, 27, 19, 7]. A critical component to all image-to-3D reconstruction methods, including traditional approaches such as multi-view stereo (MVS) [11], is to obtain camera poses for the input images. In practice, the camera information may be available from the acquisition devices, e.g., through the GPS or inertial measurement unit (IMU), while in other cases, it is estimated, e.g., using structure-from-motion (SfM) [8, 29]. In both cases, these camera poses can be noisy, thus hindering the performance of the multi-view 3D reconstruction.

In light of the importance of having accurate camera poses, various methods have been proposed to improve their estimations. One line of approaches, which can be referred to as bundle-adjusting neural fields, jointly optimize [38, 20, 12, 2, 5] camera poses along with results from rendering and geometry (e.g., depth) estimation, where the camera rays are considered *independently* for their roles in color and geometry prediction. By now, more works [3, 4, 32, 18, 16, 30, 6] realize the importance of exploiting correlations between input images, i.e., multi-view consistency, to impose additional

---

[*]Corresponding author.

38th Conference on Neural Information Processing Systems (NeurIPS 2024).

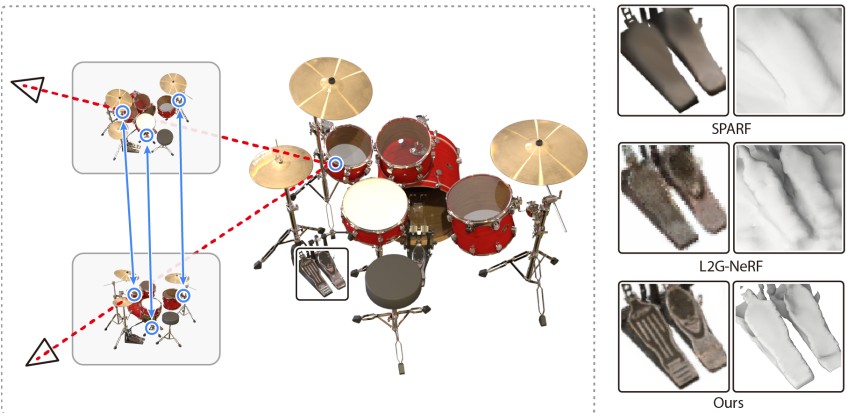

Figure 1: Our method, neural field optimization with camera ray matching (CRAYM), incorporates contextual information for per-ray processing and enforces color + geometric consistence between matched rays. Compared to SPARF [32] which utilizes dense pixel correspondences and the state-of-the-art, bundle-adjusting L2G-NeRF [5], both aimed at handling noisy camera poses, CRAYM produces superior results especially over fine details; see the zoom-ins on the right. Results are shown the Drums model from NeRF-Synthetic [26] on dense views.

constraints on the joint optimization. Along these lines, much effort has been invested into matching and optimization with respect to image features, whether convolutional or transformer-based.

Motivated by multi-view *spatial* analysis, several works have proposed geometric constraints involving camera rays and projections [16, 32, 18]. Most recently, SPARF [32] defines a *re-projection* loss as a spatial distance between image pixels to enforce that matched pixels between NeRF training images be back-projected onto the same 3D point. However, the effectiveness of this loss depends critically on how reliable the pixel correspondences are. In their work, these correspondences and their confidence estimates were both obtained by a pre-trained network [31], which is independent of the joint camera-scene optimization.

In this paper, we introduce *camera ray matching* into the joint optimization of camera poses and a neural field, referred to as a *feature volume*, which can be "probed" by the camera rays for both rendering, e.g., novel view synthesis as in NeRF [26], and 3D reconstruction, as in NeuS [34].

The key reasons for matching camera rays, instead of pixels [32, 18, 6], are two-fold. First, these rays carry 3D spatial information than just 2D pixel values to facilitate formulating *explicit* geometric losses when optimizing camera poses [16], as dictated by multi-view analysis. Second and more importantly, the camera rays can be *parameterized* by the feature volume — they carry both geometric and photometric information. Any constraint arising from camera ray matching can be passed onto the feature volume. Hence, both the matching itself and the associated matching confidence can be incorporated into the joint optimization and network training, to impose physically meaningful constraints to improve the final quality of both geometry reconstruction and rendering.

Our network, coined CRAYM (for Camera RAY Matching), takes as input an uncalibrated set of images capturing a 3D object, and is trained to predict the feature volume along with all the camera rays subjected to a combination of photometric rendering losses and geometric losses dedicated to ensuring multi-view consistency between camera rays. We consider two types of rays. The first are called *key rays*, which pass through keypoints detected in the input images, typically spanning regions with sharp features and rich textures over the 3D object. The other rays are called *auxiliary rays*, which pass through points around keypoints to offer contextual and local structural information as we reason about the key rays in our optimization framework.

As our main constraint for *matched ray coherence*, we enforce color consistency between renderings along two key rays whose corresponding keypoints from two different views are matched [9]. However, we must account for potential erroneous matches due to occlusion or unreliable local image features used by the matching network. To this end, we aggregate features along each key ray through the feature volume. The matchability between two rays is defined by a cosine similarity between the accumulated ray features and applied as a weight to either accentuate or discount the

color consistency constraint, allowing our optimization model to naturally degenerate itself to handle unrelated rays separately. Also, to improve the robustness of feature learning, we enhance the feature along each key ray by integrating features from surrounding auxiliary rays.

We evaluate our method on both the synthetic objects from NeRF-Synthetic [26] and the real scenes from UrbanScene3D [22], for novel view synthesis and 3D geometry reconstruction, over dense- and sparse-view settings. Compared to state-of-the-art alternatives, CRAYM produces superior results especially over fine details.

## 2    Related Works

**Neural Fields.**    As a pioneer work, NeRF [26] synthesizes novel views of static objects/scenes from a set of posed images by optimizing a coordinate-based neural network, which predicts the volume density and color for a sampled point in the 3D space. Since then, numerous methods have emerged to improve the rendering quality [44, 13, 1] and rendering efficiency [10, 27, 19, 7]. To extract high-quality surfaces from the learned implicit representation, NeuS [42] and VolSDF [42] propose to learn an implicit signed distance field (SDF) representation for the scenes. These methods can achieve impressive results on both novel view synthesis and 3D reconstruction, however, the requirement of precise camera pose limits their applicability in practice.

**Bundle-Adjusting Neural Fields.**    With the realization that positional encoding is susceptible to suboptimal registration, BARF [20] applies a smooth mask on the encoding at different frequency bands for a coarse-to-fine training, while [12] presents an adaptive positional encoding. L2G-NeRF [5] first learns the pixel-wise transformations for every pixel in a frame and then aligns the frame-wise transformation with the pixel-wise transformations. Common to all the above methods is that their joint optimization of pose and scene representation processes each image and each ray *separately*, without considering their multi-view correlations. As a result, the pose optimization may not be stable, thereby leading to floaters and blurriness in both novel view renderings and 3D reconstruction. Note that our method also involves per-ray processing, by combining information from auxiliary rays with that of a key ray. This is similar to the patch-level feature processing in CR-NeRF [41], which considers multiple rays indiscriminately across the image, without the notion of key rays.

**Neural Fields with Image Matching.**    Image matching can help establish geometric priors to improve the generalizability of NeRF, to either novel scenes or the sparse-view setting. MVSNeRF [3] constructs a cost volume by warping the image features extracted with a 3D CNN onto a plane sweep, from which a generalizable radiance field is learned. SparseNeuS [24] constructs a 3D volume with the variance of all the projected features from multi-view images. DBARF [4] optimizes camera poses and depth with a cost map constructed by the differences of image features. CorresNeRF [18] proposes to regularize the NeRF training with a pixel re-projection loss for the associated pixels and a depth loss for the predicted depth. GPNR [30] aggregates features of the image patches along epipolar lines with several stacked transformers. MatchNeRF [6] learns a generalizable NeRF with the cosine similarity of image features for each image pair as the shape prior. All these methods integrate image features and utilize the matching within or between different views. With more emphasis placed on multi-view geometry reasoning, SCNeRF [16] learns a pinhole model for each camera under the supervision of a re-projected ray distance loss, while SPARF [32] optimizes its network with a re-projection loss, measuring spatial distances between pixels in the same view. In contrast, the matched ray coherence formulation in our optimization accounts for both photometric and geometry information as obtained from the feature volume; the coherence constraint is also explicitly integrated into the network instead of only serving to define a loss.

## 3    Method

We are interested in neural networks that can reconstruct a 3D model, e.g., a radiance field [26] or an implicit field [34], from a set of $M$ images $\{\mathbf{I}_i\}_{i=1}^M$ capturing a 3D object from multiple views. Typically, each image is associated with a known or estimated camera pose $\mathcal{T}_i = [R_i|t_i]$, where $R_i \in \mathrm{SO}(3)$ and $t_i \in \mathrm{R}^3$. The network is trained by minimizing a photometric error $L_p$ between the input images and the multi-view renderings, $\{\hat{\mathbf{I}}_i\}_{i=1}^M$, of the target 3D object from the camera views: $\min \sum_i \sum_x ||\mathbf{I}_i(x) - \hat{\mathbf{I}}_i(x)||_2^2$, where $\mathbf{I}_i(x)$ is the color of image $\mathbf{I}_i$ at pixel $x$.

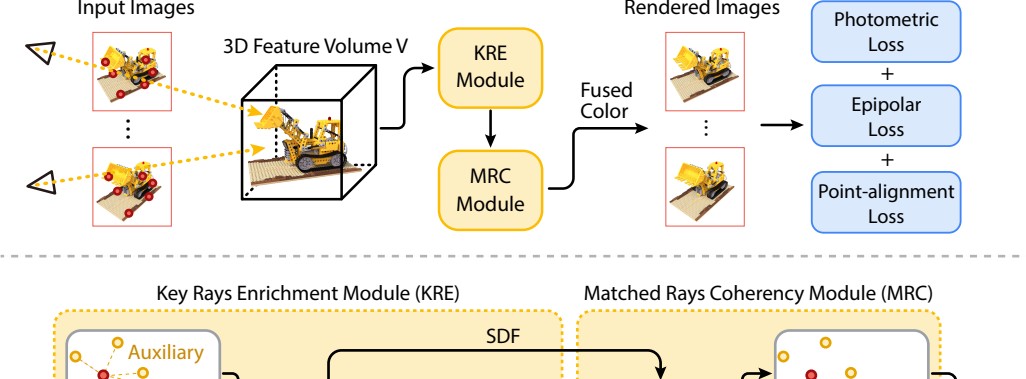

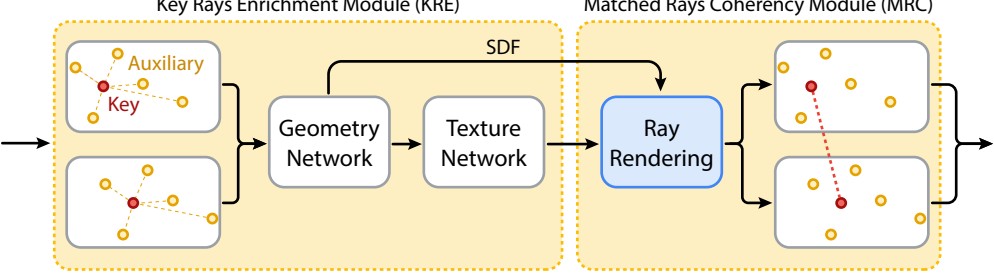

Figure 2: Overview of our CRAYM pipeline. After extracting keypoints (red dots) from input images and matching them using a pre-trained network, we train our CRAYM network to optimize a 3D feature volume $\mathcal{V}$ which encodes both geometric and photometric information about the target 3D object and can be queried by camera rays for both novel view synthesis (via the Texture Network) and 3D reconstruction (via the Geometry Network). The volume optimization is subject to photometric losses through rendering along camera rays passing through the keypoints (i.e., the key rays), which is enhanced (in the KRE) by integrating features from auxiliary rays, i.e., rays passing through nearby auxiliary points (yellow dots) in the images. Matched ray coherence (MRC) is imposed on matched key rays, in terms of color consistency, while potentially mismatched rays can be identified by comparing accumulated features along the key rays through $\mathcal{V}$. On top of the standard photometric loss, we introduce two geometric losses, the epipolar loss and point-alignment loss, to explicitly optimize ray-to-ray coherency to maximize the reconstruction quality of the feature volume.

Each pixel is associated with a specific ray in 3D from the object/scene, through the pixel center, towards the camera: $\{\mathbf{r}(t) = \mathbf{r_o} + t\mathbf{r_d} | t \geq 0\}$, where $\mathbf{r_o}$ is the camera center and $\mathbf{r_d}$ is the normalized view direction of ray $\mathbf{r}$. The rendered color of ray $\mathbf{r}$, i.e., the pixel color $\mathbf{I}_i(x)$, can be produced using volume rendering by accumulating the color and opacity $\sigma$ along the ray $\mathbf{r}$.

Considering that the camera poses can be noisy, the reconstructed radiance field or implicit field may not produce clean and sharp renderings with details. At a high noise level, some methods may even fail to produce results; see examples shown in Sections 4.2 and 4.4. Beyond existing approaches that map $\mathbf{r}(t)$ to opaque density (or opacity) and color implicitly with a ray-wise network, we propose CRAYM to learn the implicit field by matching rays across different images and formulating geometric priors.

## 3.1 The CRAYM Pipeline

Figure 2 overview our CRAYM pipeline. From the input images, our goal in the 3D neural field optimization is to construct a 3D feature volume $\mathcal{V}$ to faithfully represent the target object. In detail, we represent feature volume $\mathcal{V}$ using multi-resolution hash encoding [27] and end-to-end optimize it for the target object. The feature $f(p)$ of point $p$ in the 3D feature volume can be extracted by

$$f(p) = \mathcal{M}(\mathcal{V}(p)), \tag{1}$$

where $\mathcal{M}$ is the progressive feature mask [19] for filtering out fine-level features during early iterations of the coarse-to-fine training. Very importantly, to account for the noise in the camera poses, we parameterize the transformation matrices of the cameras as variables in the joint optimization of the pose and implicit field with the feature volume $\mathcal{V}$.

As mentioned in the introduction, we consider two types of rays to probe the feature volume, i.e., *key rays* and *auxiliary rays*. Both rays are issued from the cameras through the pixel centers. To obtain key rays $\{\mathbf{r}_k\}$, which typically associate to surface points with rich textures and sharp features, we detect keypoints on each input image using SuperPoint [9] and perform point-to-point matching between image pairs using SuperGlue [28]. Then, we can obtain a set of *sparse ray-to-ray matchings* between image pairs. Note that these results may not be accurate for various reasons such as occlusion and unreliable matching, but they provide useful information for our pipeline to start with. As for the auxiliary rays $\{\mathbf{r}_a\}$, they are sampled around the keypoints to provide contextual or local structural information when we reason about the key rays; see Section 3.2 for details.

Once optimized, the feature volume can be used for novel view synthesis or for multi-view 3D reconstruction. The color prediction for novel view synthesis is accomplished by a texture network $\Phi_t$, as in a typical NeRF [26] setting, and the latter is accomplished by a geometry network $\Phi_g$, as in a typical NeuS [34] setting. Specifically, the geometry network takes a 3D point $p$ sampled along $\mathbf{r}$ and the feature at $p$ as input to produce an SDF value and then an opaque density $\sigma$ to render the 3D object and extract the 3D reconstructions. Here, we propose the Key Rays Enrichment (KRE) module (Section 3.2) to improve the robustness in the process by enhancing the features along the key ray using the features sampled by the auxiliary rays.

Subsequently, the texture network takes the output features from geometry network, ray directions, and normal at $p$ as inputs to predict color $c(p)$ at point $p$. Further, we design the Matched Rays Coherency (MRC) module (Section 3.3) to enhance the volume rendering quality by considering matchability between rays and learning to maintain coherency between ray matchings. Particularly, the MRC module can effectively reduce the influence of mismatched rays by disambiguating the camera ray matchings.

A pair of the matched key rays, $\mathbf{r_k}$ and $\mathbf{r_k}'$, are sampled with the corresponding auxiliary rays during each iteration. The geometry network, texture network, and feature volume optimization are jointly trained end-to-end. Besides the photometric loss, we formulate the epipolar loss and point-alignment loss (Section 3.4) to explicitly promote coherency among the ray matchings and boost performance.

### 3.2 Key Rays Enrichment Module

As the input images are captured through a perspective projection, all rays in 3D through the same image should converge at a common camera point. In previous works, for each iteration, rays are optimized separately, so the pose optimization may not be stable. As different rays may back propagate gradients in different directions, the optimized poses may oscillate during the training. Hence, we introduce the KRE module to stabilize the optimization by learning structural information around each key ray. This is done by sampling auxiliary rays around the key ray to enrich the feature of the key ray with more contextual information:

$$f'(p_k) = \sum_j g(f(p_k), f(q_j)), \tag{2}$$

where $p_k$ is a point along key ray $\mathbf{r}_k$; $\{q_j\}$ are points around $p_k$ sampled along the $j$-th auxiliary ray around $\mathbf{r}_k$; and function $g$ fuses features $f(p_k)$ and $f(q_i)$. Then, we employ the geometry network $\Phi_g$ to predict the SDF value at $p_k$ and feature vector $f''(p_k)$, from which we can further obtain the color of point $p_k$ with the texture network. Please refer to the supplemental materials for the details.

### 3.3 Matched Rays Coherency Module

Next, we propose to learn the coherency of features accumulated in the 3D feature volume between the matched key rays. The purpose is to enhance the camera ray matching and account for imprecise ray matchings, since keypoints matching is performed only on local image features.

Similar to color accumulation in volume rendering, we calculate the aggregated feature along a key ray $\mathbf{r_k}$ as the feature of $\mathbf{r_k}$:

$$f(\mathbf{r_k}) = \int_0^\infty \mathcal{T}(p_k)\sigma(p_k)f''(p_k)dt. \tag{3}$$

The function $\mathcal{T}(\mathbf{r_k}(t)) = \exp(-\int_0^t \sigma(s)ds)$ denotes the accumulated transmittance along key ray $\mathbf{r_k}$.

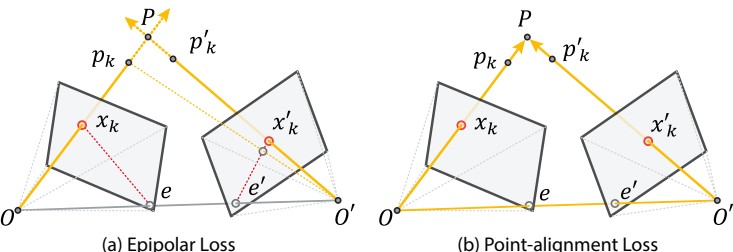

(a) Epipolar Loss          (b) Point-alignment Loss

Figure 3: Illustrating of our geometric losses. The red lines in the left subfigure are epipolar lines. The epipolar loss constrains the relative transformations between cameras, so that the projection of a keypoint $p_k$ onto the image plane of the other camera should lie on the epipolar line $e'x'_k$. With the camera poses constrained by the epipolar loss, the point-alignment loss further constrains the depth of $x_k$ and $x'_k$, aiming to align $p_k$ and $p'_k$ with $P$.

Essentially, the accumulated ray feature $f(\mathbf{r_k})$ is an integration of density-weighted features along a ray. When the network converges, the actual surface point at which ray $\mathbf{r_k}$ intersects with the paired matched key ray should have the highest density. Hence, we consider coherency between the matched rays to optimize the learning of the opacity density and point color, such that we can enhance the coherency of features accumulated along the matched key rays. In return, this will help to optimize the parameters and the 3D feature volume, when training the pipeline. Therefore, we fuse the rendered color $\mathbf{c}(\mathbf{r}_k)$ of the matched rays based on the cosine similarity between their accumulated features:

$$\mathbf{c}(\mathbf{r}_k) = w\mathbf{c}(\mathbf{r}'_k) + (1 - w)\mathbf{c}(\mathbf{r}_k), \tag{4}$$

where $w$ is the matchability calculated as the cosine distance between the accumulated features of the matched rays. When two key rays are mismatched, e.g., due to occlusion or ambiguities of weak texture areas and similar structures, our formulation can learn to degenerate itself to a form that separately optimizes individual rays.

### 3.4   Loss Function

Further, we introduce the following two geometric losses to more explicitly promote the coherency of the ray matchings:

*Epipolar loss.* Given a pair of matched keypoints $x_k$ and $x'_k$ on two different input images, which associate with camera centers $O$ and $O'$, respectively, (see the illustration in Figure 3), we can estimate the depths at $x_k$ and $x'_k$ by using a depth accumulation formulation similar to Equation 3, and then project points $x_k$ and $x'_k$ into the 3D object space to obtain 3D locations $p_k$ and $p'_k$, respectively.

If the camera poses, the matchings, and the depths are precise, the two rays through $x_k$ and $x'_k$ should precisely intersect at a common point, say $P$, on the target object surface, such that $p_k$ and $p'_k$ align with $P$. Also, we denote $e$ and $e'$ as the epipolar points on the two images; these points are the image-space locations at which the line $OO'$ intersects the two image planes; see Figure 3(a).

During the training, the depth estimation of $x_k$ can vary, so $p_k$ may vary along ray $r_k$. If the camera poses are precise, the projection of $p_k$ onto the image plane of the other camera should lie on the epipolar line $e'O'$. In case of noisy camera poses, the projection of $p_k$ may not lie exactly on $e'O'$, so we explicitly enforce the epipolarity during the training by minimizing the distance between $p_k$'s projection and the epipolar line $e'x'_k$ using

$$L_e = \frac{1}{N_k} \sum_{i=1}^{N_k} \mathrm{Dist}(Proj(p_k), e'x'_k). \tag{5}$$

Since the epipolar loss is not affected by depth, we decouple the unreliable depth estimation from the epipolar loss with ray marching to constrain the camera poses.

*Point-alignment loss.* The epipolar loss focuses on enhancing the projection consistency for producing more precise camera poses. To complement it, we introduce the point-alignment loss to facilitate depth convergence for improving the reconstruction of fine details. In detail, we consider the triangle

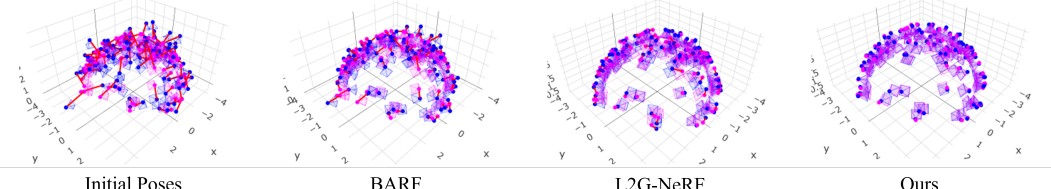

| Initial Poses | BARF | L2G-NeRF | Ours |

Figure 4: Visualization of the initial and optimized camera poses for the LEGO scene in the NeRF-Synthetic dataset [26]. (Purple: ground-truth poses; blue: initial or optimized poses; red lines: translation errors.)

formed by intersection point $P$, line segment $p_k O$, and line segment $p'_k O'$ (Figure 3(b)), and aim to minimize the distance between points $p_k$ and $p'_k$ and align them:

$$L_a = \frac{1}{N_k} \sum_{i=1}^{N_k} \text{Dist}(p_k, p'_k). \tag{6}$$

Note that depth estimation may be unreliable and likely unstable early in the training, so we use the point-alignment loss only after a certain number of training iterations.

*Overall loss.* After constructing the epipolar loss $L_e$ and the point-alignment loss $L_a$, we put them together with the photometric loss $L_p$ and SSIM loss $L_s$ to form the overall loss function

$$L = \lambda_1 * L_p + \lambda_2 * L_s + \lambda_3 * L_e + \lambda_4 * L_a, \tag{7}$$

where $L_p$ is calculated between the input images and the rendered images and is modeled as an MSE loss.

## 4 Results

We evaluate our method on the NeRF-Synthetic dataset [26] with eight synthetic objects (Section 4.2), the LLFF dataset [25], and the real scenes from the UrbanScene3D dataset [22] (Section 4.3). We compare our method on both novel view synthesis and 3D reconstruction with NeRF [26], NeuS [34], BARF [21], L2G-NeRF [5], PET-NeuS [37], SPARF [32], and BAA-NGP [23]. Since NeRF [26], NeuS [34], and PET-NeuS [37] are designed for neural implicit field with fixed and precise poses, we set the camera transformations as variables to be optimized jointly with the neural field, as in our method.

While other methods optimize the radiance field, in which the target values, radiances, of points are more independent of each other, the optimization of SDFs in NeuS and PET-NeuS poses a challenge to the requirements of non-local geometric constraints to correctly form the shape, making them more vulnerable to unstable pose optimization. As the camera rays are parameterized by our feature volume to carry both geometric and photometric information, our geometric constraints on the camera ray matching can effectively lead to better optimization of the geometry. The joint optimization of camera pose and implicit SDF may also fail to produce results for NeuS and PET-NeuS, when the camera poses are at a high noise level. With the assistance of camera ray matching, CRAYM outperforms other methods on both novel view synthesis and 3D reconstruction at varying noise levels. We report the PSNR, SSIM, and LPIPS [43] for quantitative comparisons on novel view synthesis and Chamfer distance (CD) for the 3D reconstruction. A test-time photometric pose optimization is performed to evaluate these metrics, following prior works [21, 32, 5]. The quantitative evaluations on the other metrics are provided in the supplementary materials.

### 4.1 Pose Alignment

To evaluate the registration quality of the optimized training poses, we use Procrustes analysis [14] to find the 3D similarity transformation that aligns the optimized training poses with the calibrated camera poses, following BARF [21]. As Figure 4 shows, the optimized poses produced by CRAYM align well with the ground-truth poses with lower translation errors.

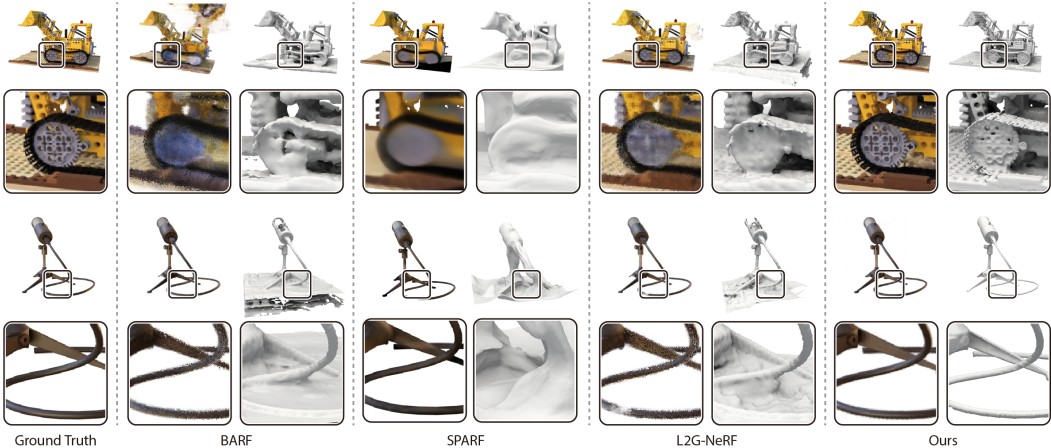

| Ground Truth | BARF | SPARF | L2G-NeRF | Ours |
| --- | --- | --- | --- | --- |

Figure 5: Qualitative comparison results of novel view synthesis and surfaces reconstruction on the synthetic objects.

The average translation errors and rotation errors are reported in Table 1. With the contextual information and feature coherency of camera ray matching, the camera poses produced by CRAYM are better optimized for the construction of the implicit filed, so that CRAYM is able to produce high-quality rendered views, as well as more precise 3D reconstructions.

Table 1: Poses registration errors evaluated on the LEGO scene in the NeRF-Synthetic dataset [26].

| Method | | Rotation (°) ↓ | Translation ↓ |
| --- | --- | --- | --- |
| BARF [21] | ICCV'21 | 9.02 | 0.81 |
| SPARF [32] | CVPR23'21 | 10.64 | 0.53 |
| L2G-NeRF [5] | CVPR23'21 | 2.90 | **0.10** |
| CRAYM (ours) | | **1.21** | 0.19 |

## 4.2 Evaluation on Synthetic Objects

For the evaluation on the NeRF-Synthetic dataset, we follow the same setting of noisy poses as L2G-NeRF [5], which perturbs the ground-truth camera poses with additive noise as the initial poses. As NeuS [34] and PET-NeuS [37] fail to produce results at such a setting, we present only the results of NeRF [26], BARF [21], SPARF [32], and L2G-NeRF [5]. The mean results of the eight objects and four of them are given in Table 2. NeRF fails to extract meshes from the reconstructed radiance fields on Hotdog and Ship. Figure 5 shows the results of view synthesis and 3D reconstruction visually for BARF, SPARF, L2G-NeRF, and our method. SPARF produces over smoothing results with dense input, as shown in Figure 5. As can be seen in Figure 5, CRAYM is able to produce clean and complete renderings and reconstructions with fewer floaters and less blurriness.

Table 2: Results on the NeRF-Synthetic dataset.

| Metrics | Method | PSNR↑ | SSIM↑ | LPIPS↓ | CD↓ |
| --- | --- | --- | --- | --- | --- |
| Chair | NeRF [26] | 16.69 | 0.77 | 0.39 | 2.23 |
| | BARF [21] | 28.55 | 0.93 | 0.06 | 0.09 |
| | SPARF [32] | 23.80 | 0.87 | 0.19 | 0.14 |
| | L2G-NeRF [5] | 30.99 | 0.95 | 0.05 | 0.10 |
| | CRAYM (ours) | **34.18** | **0.98** | **0.02** | **0.06** |
| Hotdog | NeRF [26] | 15.07 | 0.74 | 0.42 | N/A |
| | BARF [21] | 30.12 | 0.95 | 0.04 | 0.38 |
| | SPARF [32] | 29.10 | 0.93 | 0.13 | 0.07 |
| | L2G-NeRF [5] | 34.56 | 0.97 | 0.03 | 0.38 |
| | CRAYM (ours) | **36.42** | **0.98** | **0.02** | **0.05** |
| LEGO | NeRF [26] | 11.11 | 0.60 | 0.58 | 0.58 |
| | BARF [21] | 22.54 | 0.79 | 0.12 | **0.04** |
| | SPARF [32] | 22.47 | 0.80 | 0.25 | 0.09 |
| | L2G-NeRF [5] | 27.71 | 0.91 | 0.06 | 0.12 |
| | CRAYM (ours) | **31.60** | **0.96** | **0.03** | **0.04** |
| Mic | NeRF [26] | 13.08 | 0.73 | 0.53 | 0.48 |
| | BARF [21] | 30.37 | 0.96 | **0.05** | 0.28 |
| | SPARF [32] | 28.36 | 0.91 | 0.17 | 0.24 |
| | L2G-NeRF [5] | 30.91 | 0.97 | **0.05** | 0.17 |
| | CRAYM (ours) | **31.02** | **0.97** | **0.05** | **0.04** |
| Mean | NeRF [26] | 13.29 | 0.68 | 0.49 | 0.64 |
| | BARF [21] | 23.09 | 0.84 | 0.18 | 0.24 |
| | SPARF [32] | 23.90 | 0.84 | 0.23 | 0.18 |
| | L2G-NeRF [5] | 28.62 | 0.93 | 0.07 | 0.17 |
| | CRAYM (ours) | **30.34** | **0.95** | **0.05** | **0.06** |

## 4.3 Evaluation on Real Scenes

We first evaluate our method on the LLFF dataset [25] for high-fidelity view synthesis of the eight real scenes. Compared with BARF [21], L2G-NeRF [5], and BAA-NGP [23], our method is able to produce high-quality results with fewer artifacts and better scores in terms of PSNR, SSIM, and LPIPS, as shown in the Table 3.

Table 3: Qualitative comparison on novel view synthesis on the LLFF dataset [25].

| | PSNR ↑ | | | | SSIM ↑ | | | | LPIPS ↓ | | | |
| | BARF [21] | L2G-NeRF [5] | BAA-NGP [23] | CRAYM (ours) | BARF [21] | L2G-NeRF [5] | BAA-NGP [23] | CRAYM (ours) | BARF [21] | L2G-NeRF [5] | BAA-NGP [23] | CRAYM (ours) |
|---|---|---|---|---|---|---|---|---|---|---|---|---|
| Fern | 23.88 | 24.57 | 19.37 | **24.83** | 0.71 | 0.75 | 0.50 | **0.79** | 0.31 | 0.26 | 0.38 | **0.25** |
| Flower | 24.29 | 24.90 | **25.16** | 25.04 | 0.71 | 0.74 | **0.81** | 0.76 | 0.20 | 0.17 | **0.10** | 0.13 |
| Fortress | 29.06 | 29.27 | 29.24 | **29.39** | 0.82 | 0.84 | 0.83 | **0.85** | 0.13 | **0.11** | 0.14 | 0.12 |
| Horns | 23.29 | 23.12 | 19.71 | **23.30** | 0.74 | 0.74 | 0.72 | **0.75** | 0.29 | 0.26 | **0.24** | **0.24** |
| Leaves | 18.91 | 19.02 | **19.96** | 19.57 | 0.55 | 0.56 | **0.68** | 0.60 | 0.35 | 0.33 | **0.23** | 0.29 |
| Orchids | 19.46 | 19.71 | 12.45 | **19.81** | 0.57 | **0.61** | 0.14 | 0.59 | 0.29 | 0.25 | 0.42 | **0.23** |
| Room | 32.05 | 32.25 | 29.72 | **32.44** | 0.94 | **0.95** | 0.90 | 0.91 | 0.10 | **0.08** | 0.12 | 0.09 |
| T-rex | 22.92 | 23.49 | **24.56** | 23.68 | 0.78 | 0.80 | **0.86** | 0.83 | 0.20 | 0.16 | 0.11 | **0.10** |
| **Mean** | 24.23 | 24.54 | 22.52 | **24.76** | 0.73 | 0.75 | 0.68 | **0.76** | 0.23 | 0.20 | 0.22 | **0.18** |

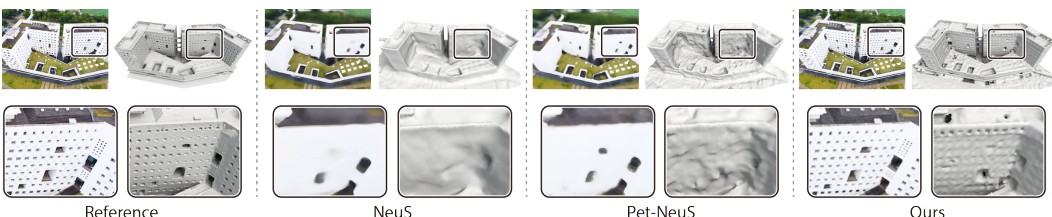

Reference      NeuS      Pet-NeuS      Ours

Figure 6: Qualitative results of novel view synthesis and surfaces reconstruction on real scenes captured by high-resolution cameras.

As the images in the LLFF dataset are captured with a restricted range of angles, we further assess our method on the real scenes PolyTech and ArtSci from the UrbanScene3D [22] dataset with two sets of drones captured images [45] associated with GPS information. The large-scale scenes are captured with hundreds of images, which share smaller overlaps than the images from the NeRF-Synthetic dataset. Considering the limitations of memory, we reduce the size of the original high-resolution images by using bicubic interpolation. As the GPS information may not be reliable and the positions in GPS may shift in meters, we preprocess the poses from GPS with COLMAP [29] and add a small noise to the calibrated poses, following L2G-NeRF [5].

Table 4: Results on the real scenes PolyTech and ArtSci in the UrbanScene3D [22] dataset.

| Scene | Method | PSNR↑ | SSIM↑ | LPIPS↓ | CD↓ |
|---|---|---|---|---|---|
| PolyTech | NeuS [34] | 14.27 | 0.42 | 0.71 | 0.10 |
| | PET-NeuS [37] | 13.64 | 0.40 | 0.75 | 0.07 |
| | CRAYM (ours) | **22.51** | **0.63** | **0.43** | **0.01** |
| ArtSci | NeuS [34] | 13.49 | 0.33 | 0.89 | 0.10 |
| | PET-NeuS [37] | 15.18 | 0.36 | 0.88 | 0.08 |
| | CRAYM (ours) | **19.37** | **0.42** | **0.63** | **0.02** |

The UrbanScene3D dataset contains high-precision LiDAR scans for the target buildings, PolyTech and ArtSCi. Therefore, we evaluate the reconstructions quality with the point cloud scans as the ground truths. We crop the reconstructed meshes according to the LiDAR scans and align them using ICP. NeRF, BARF, and L2G-NeRF produce blurry renderings and degenerated meshes for the real scenes, while SPARF may fail to process such data with dense correspondences. Therefore, we only provide the visual results of NeuS, PET-NeuS, and our method.

The quantitative results of NeuS, PET-NeuS, and our method are provided in Table 4. As Figure 6 shows, NeuS and PET-NeuS tend to produce over smoothing results, while our method is able to extract meshes with fine details. A mesh reconstructed directly from the original high-resolution images using ContextCapture[2], a commercial MVS solution, is provided as a reference.

### 4.4 Ablation Study

**Comparison on varying noise levels.** To evaluate model robustness, we evaluate on the LEGO data sample with poses at varying noise levels. The "high noise level" means we use the same noise setting as L2G-NeRF [5]; the "low noise level" means we perturb the ground-truth poses with half of the noise as L2G-NeRF; and "w/o noise" means the poses are initialized as ground-truth poses without noise. The transformation matrices of the camera poses are all set as variables to be optimized. Table 5 summarizes the results. We can see that NeuS [34] and PET-NeuS [37] are more sensitive to noise and cannot effectively handle the high-noise setting, while the other methods can not produce

---

[2]https://www.bentley.com/en/products/brands/contextcapture

accuracy reconstructions. With the contextual information learned in the camera ray matching and the explicit utilization of matched rays, CRAYM is able to obtain better results for all the noise settings.

**Ablation of major modules and losses.** To demonstrate the efficiency of the proposed modules and geometric losses, we conduct an ablation study on the LEGO data sample. The results are reported in Table 6. Similar with BARF, which applies a smooth mask on the encoding at different frequency bands for neural radiance field, we apply a progressive feature mask on the hash encoding with a coarse-to-fine training of the neural implicit field as our baseline, which combines BARF [21] and NeuS2[35]. The KRE module improves the robustness to noisy poses in the training, whereas the MRC module effectively enhances the quality of the volume renderings with ray matching. In addition to that, the proposed geometric losses further help our framework to obtain better camera pose optimization.

## 5 Conclusion and discussion

Our method, CRAYM, addresses the issue of noise camera poses for multi-view 3D reconstruction and view synthesis. The key idea is to jointly optimize a neural field and camera poses by incorporating contextual information (via KRE) and enforcing geometric and photometric consistency (via MRC and geometric losses) through camera ray matching.

Experiments demonstrate that our method outperforms state-of-the-art alternatives under various settings: dense- vs. sparse-views, and different noise levels. However, the implicit field and optimizable pose transformations may not converge when the poses are randomly initialized or extremely noisy. A stronger pose regularization prior to the field optimization may resolve this problem. Furthermore, the meshes extracted from the constructed SDFs may still contain messy inner structures over invisible areas.

Table 5: Comparing different methods at varying noise levels.

| Method | PSNR↑ | SSIM↑ | LPIPS↓ | CD↓ |
|---|---|---|---|---|
| *w/o Noise* | | | | |
| NeRF [26] | 29.08 | 0.94 | 0.04 | 0.34 |
| NeuS [34] | 21.18 | 0.82 | 0.09 | 0.04 |
| BARF [21] | 28.33 | 0.93 | 0.05 | 0.36 |
| SPARF [32] | 22.73 | 0.80 | 0.25 | 0.09 |
| PET-NeuS [37] | 21.37 | 0.82 | 0.14 | 0.10 |
| L2G-NeRF [5] | 27.94 | 0.92 | 0.06 | 0.14 |
| CRAYM (ours) | **32.72** | **0.97** | **0.02** | **0.03** |
| *Low Noise Level* | | | | |
| NeRF [26] | 24.86 | 0.88 | 0.09 | 0.34 |
| NeuS [34] | 21.76 | 0.83 | 0.14 | 0.05 |
| BARF [21] | 28.32 | 0.93 | 0.05 | 0.36 |
| SPARF [32] | 22.55 | 0.80 | 0.25 | 0.09 |
| PET-NeuS [37] | 21.34 | 0.82 | 0.11 | 0.55 |
| L2G-NeRF [5] | 27.75 | 0.92 | 0.06 | 0.21 |
| CRAYM (ours) | **32.68** | **0.97** | **0.02** | **0.04** |
| *High Noise Level* | | | | |
| NeRF [26] | 11.36 | 0.81 | 0.56 | 0.57 |
| NeuS [34] | N/A | N/A | N/A | N/A |
| BARF [21] | 14.48 | 0.69 | 0.29 | 0.04 |
| SPARF [32] | 22.47 | 0.80 | 0.25 | 0.09 |
| PET-NeuS [37] | N/A | N/A | N/A | N/A |
| L2G-NeRF [5] | 27.71 | 0.91 | 0.06 | 0.12 |
| CRAYM (ours) | **31.60** | **0.96** | **0.03** | **0.03** |

Table 6: Ablation of major modules and losses.

| Method | PSNR↑ | SSIM↑ | LPIPS↓ | CD↓ |
|---|---|---|---|---|
| L2G-NeRF [5] | 27.71 | 0.91 | 0.06 | 0.12 |
| NeuS2 [35] | 26.83 | 0.86 | 0.17 | 0.08 |
| Baseline | 27.30 | 0.91 | 0.10 | 0.06 |
| + KRE | 28.64 | 0.93 | 0.07 | 0.05 |
| + KRE + MRC | 30.41 | 0.95 | 0.04 | **0.04** |
| + $L_e$ | 29.43 | 0.92 | 0.07 | 0.05 |
| + $L_e + L_a$ | 29.95 | 0.94 | 0.06 | **0.04** |
| Our full pipeline | **31.60** | **0.96** | **0.03** | **0.04** |

A promising future work is to apply the ray matching to the 3D Gaussian splatting, which will greatly improve the rendering efficiency of CRAYM. However, extracting reconstructions with fine geometric structures from 3D Gaussians is still an open problem.

Finally, CRAYM has been designed to rely on sparse key rays for dense-view reconstruction, while a dense counterpart may bring up extra overhead. In our Matched Ray Coherency formulation, we explicitly account for potentially erroneous (i.e., low-quality) 2D matches by using the matchability between two rays as a weight to either accentuate or discount the color consistency constraint. In terms of sensitivity with respect to the density of the 2D matches, in our experiments, we have observed that even with sparse input views and sparsely distributed matched rays, CRAYM can still notably improve the optimization convergence. An effective approach to utilize ray matching for both sparse and dense inputs may further boost the performance of CRAYM.

## Acknowledgement

We thank the reviewers for their valuable comments. This work was supported in parts by NSFC (U21B2023, U2001206), ICFCRT(W2441020), Guangdong Basic and Applied Basic Research Foundation (2023B1515120026), DEGP Innovation Team (2022KCXTD025), Shenzhen Science and Technology Program (KQTD20210811090044003, RCJC20200714114435012), and Scientific Development Funds from Shenzhen University.

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

# A    Appendix / supplemental material

## A.1    Implementation

### A.1.1    Camera Pose Parameterization.

The camera poses $\{\mathbf{T}_i\}$, where $\mathbf{T}_i = [\mathbf{R}_i|\mathbf{t}_i] \in \mathrm{SE}(3)$, need to be parameterized and optimized in the training process. As a high frequency operation, pose parameterization needs to be simple and efficient, since the conversion between pose parameterization and transformation matrix is performed in every training iteration.

The position component $\mathbf{t}$ of $\mathbf{T}$ is simply represented as a three-dimensional vector. Instead of parameterizing the rotation $\mathbf{R}$ as an exponential map $\exp(\mathbf{r})$ from the Lie algebra $\mathfrak{so}(3)$ to the Lie group $\mathrm{SO}(3)$, we represent $\mathbf{R}$ as a six-dimensional vector composed by $\mathbb{R} = [v_a|v_b]$, $v_a \in \mathrm{R}^3$, $v_b \in \mathrm{R}^3$. $v_a$, $v_b$, and $v_c = v_a \times v_b$ span the three bases of the camera space. To obtain rotation matrix $\mathbf{R}$ from $\mathbb{R}$, we perform a Gram-Schmidt orthonormalization on $v_a$, $v_b$, and $v_c$, resulting in three orthonormal bases of the camera space, $\bar{v}_a$, $\bar{v}_b$, and $\bar{v}_c$. Thus, the rotation matrix is

$$R = [\bar{v}_a^T|\bar{v}_b^T|\bar{v}_c^T]. \tag{8}$$

### A.1.2    Key Rays Enrichment Module

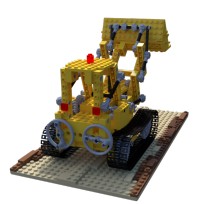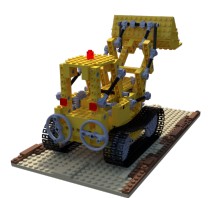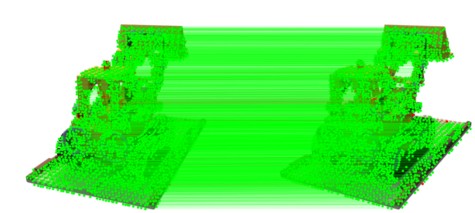

Figure 7: Visualization of the matches between two images from the LEGO scene in the NeRF-Synthetic dataset [26].

Figure 7 shows an example of matched pixels. Due to the noise in the camera poses, the surface point captured by the associated keypoint on the input image may not intersect with the key ray $\mathbf{r}_k$ exactly, the integration of features from the neighboring points $\{q_i\}$ sampled by auxiliary rays $\{\mathbf{r}_o\}$ can greatly facilitate a more stable optimization associated with key ray $\mathbf{r_k}$; see Figure 8.

Procedure-wise, we propose to first fuse the feature of $p_k$ with the features of the surrounding $\{q_i\}$ to produce the enriched feature $f'(p_k)$, which can better describe the local radiance feature and geometry feature of the captured object with structure information.

$$f'(p_k) = \sum_{i=1}^{N_o} \mathrm{Softmax}(f(p_k) * f(q_i)) * f(q_i). \tag{9}$$

The features $\{f(q_i)\}$ of the auxiliary rays $\{\mathbf{r}_a\}$ remain untouched: $f'(q_i) = f(q_i)$. Further, we adopt the geometry network $\Phi_g$ to process the features $f'(p)$ of both key rays and auxiliary rays to extract the SDF value at point $p$ in the radiance field. From the extracted SDF values, we can then produce the 3D reconstruction of the target object:

$$[\mathrm{SDF}(p)|f''(p)] = \Phi_g(f'(p)), \tag{10}$$

where $f''(p)$ is the output feature of point $p$. $f''(p)$ is one of the inputs to the texture network $\Phi_t$. The contextual information of ray matchings facilitates a more stable pose optimization and promotes the details of the geometry, i.e., the accuracy of the SDF values.

The color of point $p$ sampled on key rays or auxiliary rays can then be obtained with the texture network $\Phi_t$ as

$$c(p) = \Phi_t(f''(p), \mathbf{r_d}, \mathrm{Normal}(p)), \tag{11}$$

where $\mathbf{r_d}$ is the normalized direction of ray $\mathbf{r}$ and $\mathrm{Normal}(p)$ is the normal of the implicit surface at $p$ computed as the gradient of $\mathrm{SDF}(p)$. We take the normal at $p$ into account to boost the training of $\Phi_t$.

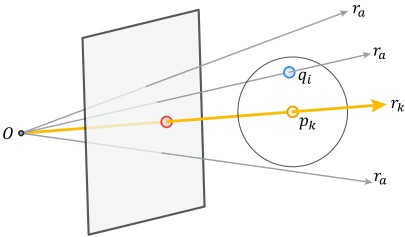

Figure 8: Illustrating the Key Ray Enrichment Module. The yellow ray $\mathbf{r}_k$ is the key ray and the gray rays $\{\mathbf{r_a}\}$ are the neighboring auxiliary rays. The surface point (the blue point) captured by the associated keypoint (the red point) on the input image may not intersect with $\mathbf{r}_k$ exactly due to the unreliable pose, while we can learn the contextual information of the surface point with the neighboring rays of point $p_k$ sampled on $\mathbf{r}_k$.

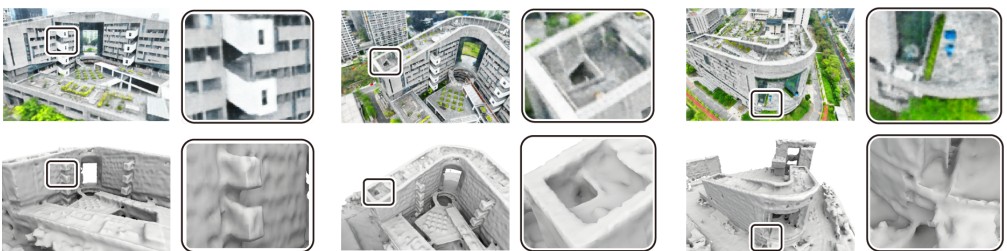

Figure 9: View synthesis and 3D reconstruction results on the real scenes CSSE produced by our method.

The color $c(\mathbf{r})$ of ray $\mathbf{r}$ is then rendered with the opaque densities and colors of all the points sampled along $\mathbf{r}$, where the opaque densities are obtained with the SDF values, normal vectors, and ray directions [34].

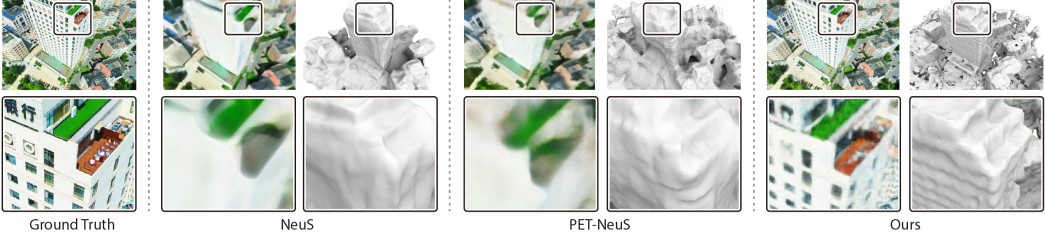

| Ground Truth | NeuS | PET-NeuS | Ours |

Figure 10: Comparison of view synthesis results on the real scenes Bank.

## A.2 Training Details

We use a 16-level hash grid [27] to encode the feature volume $\mathcal{V}$. The feature length of each level is two. Therefore, the base resolution of $\mathcal{V}$ is 32. We apply a progressive feature mask [19] on the hash encoding, which starts at level 4 and is updated to the next level every 1,000 iterations. The geometry network $\phi_g$ is implemented as a three-layer MLP with the ReLU activation for the input and hidden layers. The texture network $\phi_t$ is implemented as a four-layer MLP with the ReLU activation for the input and hidden layers. The ray directions are encoded using the spherical harmonic representation [10] and fed into the texture network $\phi_t$ together with the output features of $\phi_g$ to predict the color of the sampled points on the ray. The whole network is optimized with the AdamW optimizer with a learning rate of 0.01, $\beta = [0.9, 0.99]$, and $\epsilon = 1.0^{-15}$. The variance [34] of the geometry network $\phi_g$ is initialized as 0.3 and is optimized with a learning rate of 0.001. We adopt a warm-up training for the first 500 iterations with the LinearLR scheduler. All the experiments are conducted with an Nvidia GV100.

Table 7: Comparison of reconstruction quality on the NeRF-Synthetic [26] dataset.

| Metrics | Method | | Mean | Chair | Drums | Ficus | Hotdog | LEGO | Materials | Mic | Ship |
|---|---|---|---|---|---|---|---|---|---|---|---|
| HD↓ | NeRF [26] | ECCV'20 | 2.19 | 3.22 | 2.05 | 1.63 | N/A | 2.56 | 1.78 | 1.89 | N/A |
| | BARF [21] | ICCV'21 | 1.52 | **0.38** | 1.42 | 0.28 | 1.13 | 0.44 | 1.90 | 1.26 | 5.34 |
| | SPARF [32] | CVPR'23 | 1.11 | 0.80 | 1.42 | 1.09 | **0.54** | 0.92 | 1.35 | 1.64 | 1.10 |
| | L2G-NeRF [5] | CVPR'23 | 1.57 | 0.43 | 1.57 | 0.48 | 1.14 | 0.99 | 2.15 | 1.09 | 4.69 |
| | CRAYM (ours) | | **0.98** | **0.38** | **0.73** | **0.23** | 0.59 | **0.37** | **0.35** | **0.57** | **4.62** |
| Precision↑ | NeRF [26] | ECCV'20 | 1.17 | N/A | 1.99 | 0.71 | N/A | 3.72 | 2.12 | 0.81 | N/A |
| | BARF [21] | ICCV'21 | 15.97 | 24.60 | 15.96 | 9.23 | 7.44 | 42.89 | 18.74 | 3.279 | 5.63 |
| | SPARF [32] | CVPR'23 | 15.12 | 28.40 | 10.14 | 0.22 | 30.18 | 28.06 | 10.85 | 5.30 | 7.83 |
| | L2G-NeRF [5] | CVPR'23 | 17.36 | 21.81 | 23.01 | 10.59 | 3.38 | 43.71 | 18.31 | 10.56 | 7.54 |
| | CRAYM (ours) | | **30.59** | **33.53** | **45.44** | **17.55** | **30.50** | **49.58** | **19.16** | **38.75** | **15.60** |
| Recall↑ | NeRF [26] | ECCV'20 | 3.18 | N/A | 1.69 | 11.82 | N/A | 3.36 | 5.75 | 2.80 | N/A |
| | BARF [21] | ICCV'21 | 20.43 | 24.25 | 3.33 | 17.47 | 24.33 | 61.29 | 18.75 | 12.24 | 1.81 |
| | SPARF [32] | CVPR'23 | 19.38 | 16.17 | 6.09 | 17.54 | **37.34** | 13.68 | 12.60 | 5.40 | 19.38 |
| | L2G-NeRF [5] | CVPR'23 | 26.98 | 19.97 | 48.06 | 21.10 | 10.99 | 68.00 | 19.51 | 19.34 | 8.84 |
| | CRAYM (ours) | | **42.29** | **43.65** | **50.11** | **45.95** | 31.17 | **81.63** | **21.67** | **40.29** | **29.77** |
| F-score↑ | NeRF [26] | ECCV'20 | 1.38 | N/A | 1.83 | 1.34 | N/A | 3.53 | 3.10 | 1.25 | N/A |
| | BARF [21] | ICCV'21 | 16.32 | 24.42 | 5.51 | 12.08 | 11.39 | 50.47 | 18.75 | 5.17 | 2.74 |
| | SPARF [32] | CVPR'23 | 16.03 | 35.19 | 12.46 | 0.42 | 22.18 | 32.04 | 12.10 | 7.46 | 6.39 |
| | L2G-NeRF [5] | CVPR'23 | 20.64 | 20.85 | 31.12 | 14.10 | 5.17 | 53.22 | 18.89 | 13.66 | 8.14 |
| | CRAYM (ours) | | **34.76** | **37.93** | **47.66** | **25.40** | **30.83** | **61.69** | **20.34** | **39.50** | **20.48** |

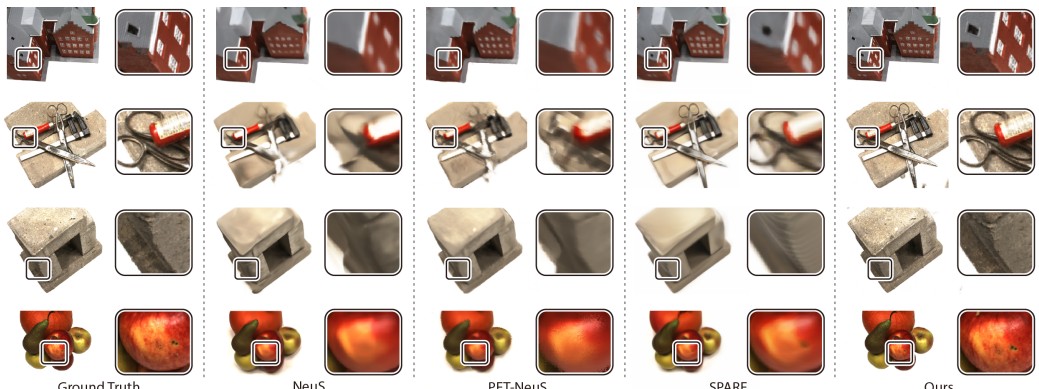

Ground Truth  NeuS  PET-NeuS  SPARF  Ours

Figure 11: View Synthesis on the DTU [15] dataset.

Table 8: Comparison of reconstruction quality on the real scenes PolyTech and ArtSci of the UrbanScene3D [22] dataset.

| Scene | Method | HD↓ | Precision↑ | Recall↑ | F-score↑ |
|---|---|---|---|---|---|
| PolyTech | NeuS [34] | 0.63 | 41.32 | 55.68 | 47.44 |
| | PET-NeuS [37] | 0.46 | 43.29 | 46.83 | 44.99 |
| | CRAYM (ours) | **0.04** | **81.59** | **91.21** | **88.13** |
| ArtSci | NeuS [34] | 0.67 | 33.73 | 31.89 | 30.24 |
| | PET-NeuS [37] | 0.32 | 21.30 | 15.69 | 18.07 |
| | CRAYM (ours) | **0.19** | **59.01** | **52.54** | **55.59** |

Table 9: Comparison of novel view synthesis on the real scene Bank from TwinTex [40].

| Method | | PSNR↑ | SSIM↑ | LPIPS↓ |
|---|---|---|---|---|
| NeuS [34] | NIPS'21 | 11.69 | 0.25 | 0.91 |
| PET-NeuS [37] | CVPR'23 | 13.22 | 0.27 | 0.93 |
| CRAYM (ours) | | **18.68** | **0.46** | **0.59** |

Table 10: Comparison of novel view synthesis on the DTU [15] dataset.

| Scene | Method | | PSNR↑ | SSIM↑ | LPIPS↓ |
|-------|--------|--|-------|-------|--------|
| Scan24 | NeuS [34] | NIPS′21 | 20.55 | 0.64 | 0.41 |
| | PET-NeuS [37] | CVPR′23 | 20.77 | 0.65 | 0.42 |
| | SPARF [32] | CVPR′23 | 22.80 | 0.78 | 0.29 |
| | CRAYM (ours) | | **30.53** | **0.96** | **0.02** |
| Scan37 | NeuS [34] | NIPS′21 | 19.27 | 0.72 | 0.30 |
| | PET-NeuS [37] | CVPR′23 | 18.68 | 0.71 | 0.30 |
| | SPARF [32] | CVPR′23 | 21.99 | 0.72 | 0.31 |
| | CRAYM (ours) | | **28.64** | **0.95** | **0.03** |
| Scan40 | NeuS [34] | NIPS′21 | 23.88 | 0.64 | 0.52 |
| | PET-NeuS [37] | CVPR′23 | 22.78 | 0.63 | 0.51 |
| | SPARF [32] | CVPR′23 | 23.85 | 0.72 | 0.38 |
| | CRAYM (ours) | | **26.86** | **0.91** | **0.06** |
| Scan55 | NeuS [34] | NIPS′21 | 16.83 | 0.66 | 0.39 |
| | PET-NeuS [37] | CVPR′23 | 23.10 | 0.73 | 0.38 |
| | SPARF [32] | CVPR′23 | 19.80 | 0.63 | 0.49 |
| | CRAYM (ours) | | **30.96** | **0.98** | **0.02** |
| Scan63 | NeuS [34] | NIPS′21 | 28.45 | 0.92 | 0.10 |
| | PET-NeuS [37] | CVPR′23 | 26.89 | 0.90 | 0.09 |
| | SPARF [32] | CVPR′23 | 27.45 | 0.92 | 0.12 |
| | CRAYM (ours) | | **31.30** | **0.98** | **0.01** |

## A.3 Reconstruction Quality

Next, we report the Hausdorff distance (HD), precision, recall, and F-score [17] for the reconstruction quality evaluated on the NeRF-Synthetic [26] dataset and the UrbanScene3D [22] dataset. The NeRF-Synthetic dataset contains eight synthetic objects, which are captured with 100 images. We evaluate our method on the two real scenes, PolyTech and ArtSci, from the UrbanScene3D [22] dataset, on which we measure the quality of the reconstructions with the high-resolution LiDAR scans as the ground truths.

Tables 7 and 8 report the reconstruction quality, compared with NeRF [26], BARF [21], SPARF [32], and L2G-NeRF [5] on the two datasets. A threshold of 0.01 is used to extract the inliers and outliers for the calculation of the precision, recall, and F1 score. The precision measures the reconstruction accuracy by calculating the distances from the reconstructed models to the ground truths. The recall measures the reconstruction completeness by determining the extent of the ground-truth points covered by the reconstructed models. A high F-score means both high accuracy and high completeness of the reconstructed models. As we can see from Tables 7 and 8, our method is able to produce high-quality reconstructions with both high accuracy and high completeness. It is worth to note that compared with other methods, our method achieves the top performance on all metrics consistently for both datasets.

## A.4 Results on Real Scenes

Further, we evaluate our method on the real scene Bank from TwinTex [40], which is a set of high-resolution drone-captured images. We preprocess the images with COLMAP [29] to obtain the calibrated poses and perturb the poses with additive noise $\xi$, where $\xi \in \mathfrak{se}(3)$ and $\xi \in \mathcal{N}(0, n\mathbf{I})$, as the initial poses, following the procedure on real scenes PolyTech and ArtSci of UrbanScene3D [22]. For these real scenes, we set $n$ as a small value 0.01.

Since TwinTex does not provide a LiDAR scan for the scene Bank, we only report the evaluation results of novel view synthesis in Table 9. Figure 10 shows the comparison on the real scene Bank visually with NeuS [34] and PET-NeuS [37]. While the other methods tend to generate renderings with blurriness, our method is able to produce sharper results with more fine details. Figure 9 further shows results of our method on another real scenes CSSE.

Table 11: Results of sparse input (3 views) on the LEGO scene from the NeRF-Synthetic dataset [26].

| Method | | PSNR↑ | SSIM↑ | LPIPS↓ | CD↓ |
|---|---|---|---|---|---|
| SPARF [32] | CVPR′23 | 15.91 | 0.69 | **0.40** | 1.27 |
| CRAYM (ours) | | **16.08** | **0.70** | 0.41 | **0.09** |

## A.5 Results on the DTU Dataset

The DTU [15] dataset is aimed at multi-view stereo (MVS) evaluation, containing image sets captured with structured light scanners mounted on an industrial robot arm. We evaluate our method on the first five image sets used in the NeuS [34], comparing with NeuS [34], PET-NeuS [37], and SPARF [32]. Each image set contains 48 images. We use 90% of them as training set and the remaining 10% images as testing data. The quantitative results of novel view synthesis on these data are shown in the Table 10. Figure 11 shows some of the novel view synthesis results visually. As we can see in Figure 11, our method produces renderings with much more details.

## A.6 Sparse Views

Since SPARF is originally designed for sparse input with re-projection loss of dense pixel correspondences, we further evaluate our method on the LEGO data with sparse input, which contains only three views as the training images. The comparison of SPARF [32] and our method is shown in Table 11. Though CRAYM aims for bundle-adjusting neural implicit field with dense views as input, it still obtains a result comparable with SPARF, for sparse input views.

