# OpenReview forum: "CRAYM: Neural Field Optimization via Camera RAY Matching"
_NeurIPS.cc/2024/Conference — NeurIPS 2024 poster_

### Official Review · Reviewer_Bu9t · 2024-07-05

**Soundness:** 3
**Presentation:** 4
**Contribution:** 2
**Rating:** 5
**Confidence:** 3

**Summary:**

The manuscript #3263 entitled "CRAYM: Neural Field Optimization via Camera RAY Matching" proposes a novel uncalibrated NeRF strategy based on prior keypoints matching across images. Specifically, the authors propose two novelties to improve the quality of the reconstruction and the pose estimation of the cameras: 1) Enriched ray features using surrounding rays sampled around the keypoints and 2) a ray matching index which can be used to re-weight the color regression part, leading to better robustness to occlusions.
The proposed technique has been evaluated across various standard datasets and against meaningful NeRF-like algorithms.

**Strengths:**

- The idea of separating key rays and auxiliary rays is interesting and meaningful.
- Numerous and conclusive results.
- Assessment on a large number of datasets.
- Good ablation study underlying the benefit of each novelty.

**Weaknesses:**

- The robustness of the approach against outlier matches is not evaluated. Introducing artificial outliers (wrongly matched keypoints) into the dataset to assess how well the technique can handle mismatches would be of some interest.
*Question*: Would the matched rays consistency and the epipolar geometry compensate for that? Or would the training diverge?

- As stated in the literature review of this manuscript, other approaches taking advantage of the epipolar geometry and prior matching have already been designed in this context. I have difficulty understanding what is significantly different with this work apart from the sampling of additional surrounding rays and the matching ray index used to weight color prediction using image pairs. These two novelties seem rather incremental, but they nonetheless lead to strongly improved results.
*Question*: I assume that the other keypoints-based approaches are not "self-calibrated". Is the proposed technique the first "keypoint-based" calibration-free NeRF? If it is not the case, it would be meaningful to compare against such techniques too.

- Adding surrounding rays around a key ray appears to be quite effective; however, the sampling of auxiliary rays is not well described in the paper.
*Question*: How are the rays sampled?

- The initialization of the pose lacks details.
 *Question*: What is the effect of the pose initialization on the result?

- The intrinsic parameters of the camera could additionally be optimized.
*Question*: Just out of curiosity, have you conducted such an experiment?

- In equation (4), it seems that the proposed solution considers only pairs of images.
*Question*: How are those pairs selected?

The proposed approach is inspired by existing techniques integrating matched keypoints (like using the epipolar loss) and other techniques, such as NeuS.

- The loss function contains many regularization factors.
*Question*: Is the final loss hard to balance?


Overall, the paper is interesting and proposes a few contributions that seem to lead to strongly improved results. Moreover, the approach has been evaluated on various standard datasets and against representative methods. However, this novel approach remains relatively incremental, and many points remain to be clarified regarding the robustness of the technique. For all the above-mentioned reasons, I would like to issue a rather mixed opinion regarding the acceptance of this work for this conference.

**Questions:**

The questions are integrated in the Weaknesses part.

**Limitations:**

The limitations of the paper may not have been entirely investigated, specifically in terms of robustness. For instance, the influence of the initial pose and outliers has not been demonstrated.

---

> ### Author Rebuttal · Authors · 2024-08-06
>
> Thank you for the insightful comments and suggestions, especially the series of detailed questions, which we will all answer below. We hope the answers will address your concerns.
>
> Q: The robustness of the approach against outlier matches is not evaluated. Introducing artificial outliers (wrongly matched key points) into the dataset to assess how well the technique can handle mismatches would be of some interest. Would the matched rays' consistency and the epipolar geometry compensate for that? Or would the training diverge?
>
> A: Yes, our Matched Ray Coherency formulation is precisely designed to compensate for wrongly-matched rays. The use of auxiliary rays serves to improve the robustness of ray matching. In practice, erroneous keypoint/ray matches do happen and they probably do not need to be "artificially introduced." For example, when the noise level increases, wrongly-matched rays will tend to happen more frequently. To this end, our robustness experiments against different noise levels (see Table 2 in the paper) can be regarded also as a robustness test on our method's ability to handle wrongly-matched rays.
>
> Q: As stated in the literature review of this manuscript, other approaches taking advantage of the epipolar geometry and prior matching have already been designed in this context. I have difficulty understanding what is significantly different with this work apart from the sampling of additional surrounding rays and the matching ray index used to weight color prediction using image pairs. These two novelties seem rather incremental, but they nonetheless lead to strongly improved results.  _Question_: I assume that the other keypoints-based approaches are not "self-calibrated". Is the proposed technique the first "keypoint-based" calibration-free NeRF? If it is not the case, it would be meaningful to compare against such techniques too.
>
> A: Yes, CRAYM is the first "keypoint-based" calibration-free method, as far as we are aware of.  Other approaches usually compute the reprojection distance of the matched pixels, while we compute the distance between the reprojected pixel and the epipolar line as the epipolar loss, thereby boosting the convergence of the camera poses.
>
> Q: Adding surrounding rays around a key ray appears to be quite effective; however, the sampling of auxiliary rays is not well described in the paper.  _Question_: How are the rays sampled?
>
> A: Currently, we randomly sample n auxiliary rays from the k nearest rays of key rays. More details of auxiliary rays will be given in the revision.
>
> Q: The initialization of the pose lacks details.  _Question_: What is the effect of the pose initialization on the result?
>
> A: As stated in the paper, we follow L2G-NeRF to perturb the ground-truth camera poses with additive noise as the initial poses.  Table 2 shows the results of using different noise levels on the initial poses. In general, well-initialized poses produce robust and fast convergence.
>
> Q: The intrinsic parameters of the camera could additionally be optimized.  _Question_: Just out of curiosity, have you conducted such an experiment?
>
> A: All the evaluations assume known camera intrinsics, as most of the other methods do. We have performed experiments to additionally optimize the intrinsics and they succeed in learning the intrinsics with some more interactions.
>
> Q: In equation (4), it seems that the proposed solution considers only pairs of images.  _Question_: How are those pairs selected?
>
> A: We do not select image pairs for training. Instead, we randomly select a ray from the ray set generated with all the training images and their poses. If a matched ray is found for it, the matched ray pair is optimized. Otherwise, the selected ray is optimized on its own.
>
> Q: The loss function contains many regularization factors.  _Question_: Is the final loss hard to balance?
>
> A: Empirically, we found that these items in the loss function promote one another rather than weakening/antagonizing one another and we do not need to tune them for an optimal balance in producing our results.

---

> > ### Comment · Reviewer_Bu9t · 2024-08-13
> >
> > I would like to sincerely thank the authors for their responses, which have clarified many of my questions. However, in light of the other reviews, I agree that this paper appears to be rather incremental. For this reason, I would like to maintain my original rating of BA.

---

> > > ### Author Response · Authors · 2024-08-14
> > >
> > > Dear Reviewer Bu9t,
> > >
> > > Thank you for your thorough reviews and thoughtful evaluation of our paper. We greatly appreciate your positive rating. We will continue to refine our manuscript in light of your feedback and hope to meet your expectations in the final version.

---

### Official Review · Reviewer_67PC · 2024-07-12

**Soundness:** 2
**Presentation:** 2
**Contribution:** 3
**Rating:** 5
**Confidence:** 5

**Summary:**

This paper presents a new technique called camera ray matching, which is integrated into the joint optimization of camera poses and a neural field. The method utilizes an uncalibrated set of images as input, incorporating photometric and geometric constraints through key points and key rays matching, with the aim of enhancing the quality of novel view rendering and 3D surface reconstruction. The approach comprises two simple modules and is implemented using grid-based representation (iNGP). Photometric experiments were exclusively compared with MLP-based methods, specifically NeRF-like, on the synthetic dataset of the vanilla NeRF, while geometric experiments demonstrate some positive results. The authors provide additional results in the appendix.

**Strengths:**

This work is a positive extension to the field of neural reconstruction (like NeRF and SDF) under the setting of images captured with noisy poses. Authors make efforts to simultaneously solve the problems involving the camera pose, detailed renderings, and accurate surface reconstruction. Experiments show good results.

**Weaknesses:**

Too many factors are taken into account in the writing simultaneously, which leads to a lack of clear theme or a clear academic or technical problem to be addressed in this paper. The work appears to build incrementally upon previous research and offers limited novelty. The so-called Epipolar loss and Point-alignment loss are actually based on Bundle Adjustment (BA), using key points matching, which has been previously applied in the optimization of neural reconstruction in works such as SCNeRF, BARF, L2G, and Level2sfm. The proposed two modules do not bring significant innovation. It is also confusing that this work is implemented using a grid-based representation (i.e., iNGP), while the compared methods are implemented using MLP-based representation, which does not allow for a precise and fair comparison. I suggest that the authors refer to ZipNeRF for guidance on how to formulate research problems and conduct appropriate comparisons.

**Questions:**

Based on the mentioned shortcomings, I have several questions:

1. What is the main research problem addressed in this paper? It appears that the paper intends to address three problems simultaneously: camera pose, fine-detailed rendering, and accurate surface reconstruction. However, if the goal is to solve the camera pose problem, it would be beneficial to present more results related to the accurate regression of camera poses before discussing high-fidelity results. Could you provide additional results about the accurate regression of camera poses?

2. If the primary focus is on the issue of high-fidelity rendering under noisy camera poses, the experimental results do not offer strong conviction. It would be beneficial to showcase more experimental results on different types of datasets, similar to how BARF, SPARF, and L2G-NeRF were compared on the LLFF dataset, with an emphasis on trajectory of pose optimization and high-fidelity renderings.

3. Initialization of camera poses is a sensitive issue in joint optimization. Have you attempted to change the initial camera poses or initialize poses using COLMAP to test the robustness of pose regression?

4. This work is implemented using grid-based representation, while the compared methods are implemented using MLP-based representation, leading to an imprecise and unfair comparison. I am curious about the total number of iterations each method was trained for. BARF and L2G-NeRF trained all models for 200K iterations, and SPARF trained models for 100K iterations. It seems likely that this paper trained for significantly fewer iterations (perhaps 10k iterations) due to the inherent faster convergence of the grid-based representation. It would be fairer to re-implement this paper using MLP-based representation and train for a duration comparable to other methods, then make comparisons. Additional experiments are needed to support these claims.

5. The comparison under sparse-view conditions lacks precision and distinction. SPARF evaluated on the DTU dataset with only 3 views, while you used 48 images in your setting, although you have reported results for sparse input (3 views) on the LEGO data, it may not be sufficiently convincing.

6. Since Neus and PET-NeuS have compared the surface reconstructions on the DTU dataset, I would appreciate the inclusion of visual results. Additional ablation studies on 3D surface reconstruction using your modules on the DTU dataset would enhance the paper.

7. Considering the integration of multi-resolution hash encodings into a neural surface representation, I recommend that the authors compare with Neus2. Neus2 has compared their work with Instant-NGP and Neus, showcasing fast training and detailed surface reconstruction. If the goal is to demonstrate the superiority of 3D surface reconstruction, it is most appropriate to consider Neus2 in the comparison.

**Limitations:**

The authors have identified a limitation where the meshes extracted from the constructed SDFs may still contain messy inner structures over invisible areas. I recommend that the authors explore the possibility of finding the SDFs of surface points instead of the SDFs of each sampled point along rays. This would involve assessing whether the depth accumulated by all points along the rays is more accurate than the output surface generated by all sampled points.

---

> ### Author Rebuttal · Authors · 2024-08-06
>
> Thank you for the careful review and insightful questions. The reviewer's various suggestions are right on and we hope the rebuttal will alleviate the concerns raised.
>
> Q: Additional results on accurate regression of camera poses?
>
> A: Please refer to Table 9 in the supplemental material.  It shows *exactly* the quality of pose regression by our method. The results show that CRAYM yields better pose estimation than other methods. If requested and space permitting, we can move the relevant materials to Section 4 of the main paper.
>
> Q: It would be beneficial to showcase more experimental results on different types of datasets with an emphasis on trajectory of pose optimization and high-fidelity renderings.
>
> A: This request coincides with what Reviewer GtQM asked above. Hence, we copy the results on the LLFF dataset below. The trajectories of LEGO from NeRF-Synthetic (Blender) are shown in the uploaded PDF file. Other trajectories will be provided in the revised paper.
>
> Table 1: Mean scores over three metrics.
>
> |                        | BARF  | L2G-NeRF | BAA-NGP | CRAYM (ours)  |
> | ---------------- | ------- | ------------ | ------------ | ----------------- |
> | PSNR Mean    | 24.23 |         24.54 |         22.52 |           **24.76** |
> | SSIM Mean     |   0.73 |           0.75 |           0.68 |             **0.76** |
> | LPIPS Mean    |   0.23 |           0.20 |           0.22 |             **0.18** |
>
> Table 2: PSNR scores over various models.
>
> | PSNR     | BARF  | L2G-NeRF | BAA-NGP   | CRAYM     |
> | -------- | ----- | -------- | --------- | --------- |
> | Fern     | 23.88 | 24.57    | 19.37     | **24.83** |
> | Flower   | 24.29 | 24.90    | **25.16**     | 25.04 |
> | Fortress | 29.06 | 29.27    | 29.24     | **29.39** |
> | Horns    | 23.29 | 23.12    | 19.71     | **23.30** |
> | Leaves   | 18.91 | 19.02    | **19.96**     | 19.57 |
> | Orchids  | 19.46 | 19.71    | 12.45     | **19.81** |
> | Room     | 32.05 | 32.25    | 29.72     | **32.44** |
> | T-rex    | 22.92 | 23.49    | **24.56** | 23.68     |
> | Mean     | 24.23 | 24.54    | 22.52     | **24.76** |
>
> Table 3: SSIM scores over various models.
>
> | SSIM     | BARF | L2G-NeRF | BAA-NGP  | CRAYM    |
> | -------- | ---- | -------- | -------- | -------- |
> | Fern     | 0.71 | 0.75     | 0.50     | **0.79** |
> | Flower   | 0.71 | 0.74     | **0.81** | 0.76     |
> | Fortress | 0.82 | 0.84     | 0.83     | **0.85** |
> | Horns    | 0.74 | 0.74     | 0.72     | **0.75** |
> | Leaves   | 0.55 | 0.56     | **0.68** | 0.60     |
> | Orchids  | 0.57 | **0.61** | 0.14     | 0.59     |
> | Room     | 0.94 | **0.95** | 0.90     | 0.91     |
> | T-rex    | 0.78 | 0.80     | **0.86** | 0.83     |
> | Mean     | 0.73 | 0.75     | 0.68     | **0.76** |
>
> Table 4: LPIPS scores over various models.
>
> | LPIPS    | BARF | L2G-NeRF | BAA-NGP  | CRAYM    |
> | -------- | ---- | -------- | -------- | -------- |
> | Fern     | 0.31 | 0.26     | 0.38     | **0.25** |
> | Flower   | 0.20 | 0.17     | **0.10** | 0.13     |
> | Fortress | 0.13 | **0.11** | 0.14     | 0.12     |
> | Horns    | 0.29 | 0.26     | **0.24** | **0.24** |
> | Leaves   | 0.35 | 0.33     | **0.23** | 0.29     |
> | Orchids  | 0.29 | 0.25     | 0.42     | **0.23** |
> | Room     | 0.10 | **0.08**     | 0.12     | 0.09 |
> | T-rex    | 0.20 | 0.16     | 0.11     | **0.10** |
> | Mean     | 0.23 | 0.20     | 0.22     | **0.18** |
>
> Q: This work is implemented using grid-based representation, while the compared methods are implemented using MLP-based representation, leading to an imprecise and unfair comparison.
>
> A: This is a fair point when considering the *original* implementations. However, when making comparisons to CRAYM, we did implement BARF with *grid-based* representations as the baseline and the comparison results are shown in paper and also in the four tables above. The gap between the BARF baseline (27.30) and CRAYM (31.60) shows that the improvement comes mainly from exploiting the matched rays in our proposed formulation. The results of another grid-based representation method, namely NeuS2, are also provided in the table at the end of this rebuttal.
>
> Q: The comparison under sparse-view conditions lacks precision and distinction. SPARF was evaluated on the DTU dataset with only 3 views, while you used 48 images in your setting, although you have reported results for sparse input (3 views) on the LEGO data, it may not be sufficiently convincing.
>
> A: The overall synthesis quality with sparse views is demonstrated with the whole evaluation image set.  We randomly select three images from the evaluation set as the final results. The comparison with SPARF is shown below, with our method outperforming:
>
> |       | PSNR      | SSIM     | LPIPS    | CD        |
> | ----- | --------- | -------- | -------- | --------- |
> | SPARF | 15.91     | 0.69     | 0.40     | 1.270     |
> | CRAYM | **16.08** | **0.70** | **0.41** | **0.094** |
>
> Q: I recommend that the authors compare with Neus2.
>
> A: Good idea. Please first note that in Table 4 in the submitted paper, we applied a progressive feature mask on the multi-resolution hash encoding into a neural surface representation as our baseline, which combines BARF and NeuS2. The results of NeuS2, the BARF baseline, l2G-NeRF, and CRAYM are shown in the table below, with our method coming on top. Although NeuS2 produces better reconstructions than L2G-NeRF, L2G-NeRF outperforms NeuS2 in the quality of the rendered views.
>
> |          | PSNR      | SSIM     | LPIPS    | CD        |
> | -------- | --------- | -------- | -------- | --------- |
> | NeuS2    | 26.83     | 0.86     | 0.17     | 0.075     |
> | BARF Baseline | 27.30     | 0.91     | 0.10     | 0.063     |
> | L2G-NeRF | 27.71     | 0.91     | 0.06     | 0.115     |
> | CRAYM    | **31.60** | **0.96** | **0.03** | **0.039** |

---

> > ### Author Response · Authors · 2024-08-14
> >
> > Dear Reviewer 67PC,
> >
> > Thank you for your comprehensive reviews and constructive suggestions. We have evaluated our method on the LLFF dataset and compared it with NeuS2. The quality of pose regression by our method is presented in Table 9 of the supplemental material. We hope the rebuttal will help address your concerns, and we are eager to resolve any further issues to ensure our submission meets the high standards of the conference.

---

> ### Comment · Reviewer_67PC · 2024-08-14
>
> Thanks for the great efforts of the authors! After carefully reading the responses both from the authors and other reviewers, most of my concerns have been addressed. I tend to improve my rating to BA now and suggest the authors add more experimental details/results and discussions in the revision.

---

> > ### Author Response · Authors · 2024-08-14
> >
> > Dear Reviewer 67PC,
> >
> > Thank you very much for recognizing our work and for your positive rating. We appreciate your feedback and will continue to refine our manuscript. In the revision, we will include optimized trajectories, a discussion on grid-based methods, additional experimental details and results.
> >
> > Thanks again,
> > The authors

---

### Official Review · Reviewer_LukG · 2024-07-12

**Soundness:** 4
**Presentation:** 4
**Contribution:** 4
**Rating:** 8
**Confidence:** 5

**Summary:**

This work suggests a novel neural representation and training scheme that jointly solves for the scene representation and the multi-view camera localization. It is done using several new ideas that generalize existing NeRF based methods.
The representation itself is a combination of a geometry-network, which predicts a signed-distance-function (SDF) and a feature vector, that are fed into the texture-network that predicts the usual color and density values.
The main key novelty, is that the optimization is done over matching rays, obtained from matching keypoints using a pretrained network. The standard photometric loss function is extended to incorporate an epipolar loss (that constrains the camera positions) and a point-alignment loss that ensures the ray intersect at the predicted depth estimates along the rays. Another strong addition, is the use of 'auxiliary' rays around each matched pair of rays, from which features are fused to produce a more robust representation, that can aid the optimization under errors in matching and camera poses.
Extensive experiments demonstrate the importance of each component and the strong performance of their combination.

**Strengths:**

* The paper presents an extension of the NeRF framework, based on several novel and interesting additions that are framed in a single pipeline. The experimental results show that these contributions work well together and yield new state-of-the-art results, across the board.
* One promising idea, in my view, is the joint optimization of both geometry and texture networks, which clearly complement eachother and are helpful in obtaining stronger and more accurate constraints on the scene understanding (as opposed to most NeRF pipelines that focus on image reconstruction and are less accurate for 3D reconstruction).
* The other strong idea, is the joint optimization of matching rays, once again - imposing consistency contraints (on both camera and surface locations) that were not previously exploited to such an extent in prior work.
* The paper is well written and the contributions are very clearly highlighted, while the understanding of the conventional parts is left for the reader (which is mostly fine).

**Weaknesses:**

* Reproducibility - I believe that many details are missing (including from the appendix) for one to be able to implement the proposed method. For example:
   * What are the settings of the preprocessing SuperPoint and SuperGlue matching? What is the typical match density?
   * How are the auxiliary rays sampled? How many and under which distribution?
   * What is the function g in Eq 2 that fuses the key and auxiliary features?
   * What are the balance weights in the final loss (Eq. 7)?
   * How are poses initialized?
* Complexity - There is no discussion what so ever about the impact of the suggested changes on memory and runtime complexity, but at traning and in inference.
* Qualitative results are relatively limited.
   * Synthesized images are all very small, so it is difficult to appreciate the fidelity.
   * No depth images are shown
   * No examples of key and auxiliary point matches are shown (over entire images)

**Questions:**

* Is the training done only on matching key points? If so, what happens if correct matching keypoints do not 'cover' the space adequately?
* How sensitive is the method to the quality and density of the 2D matches? It reportedly worked well on sparse image sets, which is somewhat surprising.
* What are the runtimes (training and inference) compared to some baseline methods?
* How different is the use of auxiliary rays compared to sampling from conical frustums as in Mip-NeRF?

**Limitations:**

Adequately discussed.

---

> ### Author Rebuttal · Authors · 2024-08-06
>
> Thank you for your care and insights in the reviews and the encouraging remarks!
>
> Q: Reproducibility
>
> A: All the implementation details mentioned along with other useful information will be added to the supplemental material in the revision. The source code and any data used will surely be released upon paper publication.
>
> Q: Complexity
>
> A: When training CRAYM, each iteration consumes around twice the amount of resources vs. other low-resource alternatives, since our method optimizes two matched rays together in each iteration.  However, CRAYM converges much faster than these alternatives (10-20k iterations vs. 200k iterations) during training. During inference, the views are synthesized from the optimized field with the same rendering process and each ray is rendered separately, so the inference efficiency of CRAYM is the same as the other methods.
>
> Q: Qualitative results, synthesized images, depth images, example key, and auxiliary point.
>
> A: We will provide these results in the revised paper as space would allow. All other requested additions will be incorporated into the supplementary material.
>
> Q: Is the training done only on matching key points?
>
> A: No. Matched rays are optimized together, whereas rays without matched companions are optimized separately. Without using any matched rays, CRAYM will be downgraded to the baseline method.
>
> Q: How sensitive is the method to the quality and density of the 2D matches?
>
> A: In our Matched Ray Coherency formulation, we explicitly account for potentially erroneous (i.e., low-quality) 2D matches by using the matchability between two rays as a weight to either accentuate or discount the color consistency constraint; see line 66 in the paper. In terms of sensitivity with respect to the density of the 2D matches, in our experiments, we have observed that even with sparse input views and sparsely distributed matched rays, CRAYM can still notably improve the optimization convergence. We would be happy to provide such a sensitivity analysis in the revision.

---

### Official Review · Reviewer_GtQM · 2024-07-15

**Soundness:** 2
**Presentation:** 2
**Contribution:** 2
**Rating:** 5
**Confidence:** 4

**Summary:**

This paper introduces Camera Ray Matching for optimizing camera poses and neural fields from multi-view images. The optimized feature volume supports novel view synthesis and 3D geometry reconstruction by probing camera rays, which carry both geometric and photometric information. CRAYM claims to improves efficiency and accuracy by focusing on keypoints and integrating multi-view consistencies, enhancing both geometric reconstruction and photorealistic rendering. The method shows result in NVS and geometry reconstruction compared to baseline methods.

**Strengths:**

- The paper is well-structured and easy to follow.

**Weaknesses:**

- Experiments were only conducted on NeRF-synthetic datasets and not on LLFF datasets.
- Comparison is made with older baseline methods (e.g., SPARF, BARF, L2G) which are more than 2 years old. It’s recommended to include more recent methods such as NoPe-NeRF and BAA-NGP.
- It is suggested that the authors perform Neural Image Alignment to enhance the evaluation.

**Questions:**

Figure 5 and Table 3 appear to have a significant overlap in the information they present.

**Limitations:**

YES

---

> ### Author Rebuttal · Authors · 2024-08-06
>
> We appreciate the suggestions in the review and hope the rebuttal will help address the concerns raised.
>
> Q: Experiments were only conducted on NeRF-synthetic datasets and not on LLFF datasets. Also add Neural Image Alignment to enhance the evaluation.
>
> A: We have evaluated our method on the LLFF dataset and the results are listed in the four tables below. The first table shows the mean scores for PSNR, SSIM, and LPIPS, comparing our method (CRAYM) to three baselines including BAA-NGP, a recent method suggested by the reviewer. Our method outperforms all the baselines over all metrics. The remaining three tables show performance numbers for each of the metrics on various models.
>
> Evaluations on the Neural Image Alignment task will be added in the revision.
>
> Table 1: Mean scores over three metrics.
>
> |                        | BARF  | L2G-NeRF | BAA-NGP | CRAYM (ours)  |
> | ---------------- | ------- | ------------ | ------------ | ----------------- |
> | PSNR Mean    | 24.23 |         24.54 |         22.52 |           **24.76** |
> | SSIM Mean     |   0.73 |           0.75 |           0.68 |             **0.76** |
> | LPIPS Mean    |   0.23 |           0.20 |           0.22 |             **0.18** |
>
> Table 2: PSNR scores over various models.
>
> | PSNR     | BARF  | L2G-NeRF | BAA-NGP   | CRAYM (ours) |
> | ---------- | ------- | ------------ | ------------- | ----------------- |
> | Fern       |   23.88 |        24.57 |      19.37     |           **24.83** |
> | Flower   |    24.29 |       24.90 |      **25.16**     |           25.04 |
> | Fortress |    29.06 |      29.27 |      29.24      |           **29.39** |
> | Horns    |    23.29 |      23.12 |       19.71     |           **23.30** |
> | Leaves  |    18.91 |     19.02  |       **19.96**     |           19.57 |
> | Orchids |    19.46 |     19.71  |       12.45     |           **19.81** |
> | Room    |    32.05 |     32.25  |       29.72     |           **32.44** |
> | T-rex      |   22.92 |     23.49   |    **24.56**  |             23.68     |
> | Mean     |   24.23 |     24.54   |      22.52     |            **24.76** |
>
> Table 3: SSIM scores over various models.
>
> | SSIM      | BARF  | L2G-NeRF | BAA-NGP   | CRAYM (ours) |
> | ---------- | ------- | ------------ | ------------- | ----------------- |
> | Fern       |    0.71 |       0.75     |        0.50     |             **0.79** |
> | Flower    |   0.71 |        0.74     |       **0.81** |             0.76     |
> | Fortress  |   0.82 |       0.84     |        0.83     |             **0.85** |
> | Horns     |   0.74 |       0.74     |        0.72     |             **0.75** |
> | Leaves   |   0.55 |       0.56     |       **0.68** |              0.60     |
> | Orchids  |   0.57 |      **0.61** |        0.14     |              0.59     |
> | Room     |   0.94 |      **0.95** |        0.90     |              0.91     |
> | T-rex      |   0.78 |        0.80     |       **0.86** |              0.83     |
> | Mean     |   0.73 |        0.75     |        0.68     |             **0.76** |
>
> Table 4: LPIPS scores over various models.
>
> | LPIPS     | BARF  | L2G-NeRF | BAA-NGP   | CRAYM (ours) |
> | ---------- | ------- | ------------ | ------------- | ----------------- |
> | Fern       |     0.31 |      0.26     |        0.38     |             **0.25** |
> | Flower   |     0.20 |      0.17     |       **0.10** |               0.13     |
> | Fortress |     0.13 |     **0.11** |        0.14     |               0.12     |
> | Horns    |     0.29 |      0.26     |       **0.24** |              **0.24** |
> | Leaves  |     0.35 |      0.33     |       **0.23** |               0.29     |
> | Orchids |     0.29 |      0.25     |        0.42     |              **0.23** |
> | Room    |     0.10 |      **0.08**    |        0.12     |              0.09 |
> | T-rex      |     0.20 |      0.16     |        0.11     |              **0.10** |
> | Mean     |     0.23 |      0.20     |        0.22     |              **0.18** |
>
> Q: Include more recent methods such as NoPe-NeRF and BAA-NGP.
>
> A: The comparisons with BAA-NGP were included in the tables above, and as shown, our method outperforms all the baselines including BAA-NGP, over all three metrics PSNR, SSIM, and LPIPS, on the LLFF dataset.
>
> As for NoPe-NeRF, we had tested the code released by the authors of that work. However, since NoPe-NeRF heavily relies on depth estimation results to adjust the camera poses, it cannot effectively generalize to other datasets, as in our setting.
>
> Q: Figure 5 and Table 3.
>
> A: Thanks! We will reduce the overlap in the revision.

---

> > ### Comment · Reviewer_GtQM · 2024-08-13
> >
> > The author addresses most of my concerns. Please add a discussion about NoPe-NeRF, even if not making a direct comparison. I am raising the score to BA.

---

> > > ### Author Response · Authors · 2024-08-14
> > >
> > > Dear Reviewer GtQM,
> > >
> > > We are pleased to learn that the majority of your concerns have been addressed. Thank you for your thoughtful and constructive feedback on our submission. We will include a discussion about NoPe-NeRF in the revised version. We are committed to addressing any additional concerns you may have to ensure our paper meets the high standards of the conference.

---

### Author Rebuttal · Authors · 2024-08-07

We want to thank all the reviewers for their comprehensive reviews of our paper. The insightful questions and various suggestions for additional experiments and clarifications will surely strengthen this work.

Here, let us first start with some quick remarks on common reviewer comments. The individual rebuttals will fully address all the reviewer questions and concerns, with additional experiment results provided as requested.

Two reviewers raised questions about the robustness/sensitivity of our method against wrongly-matched keypoints/rays. To this, let us reiterate that our Matched Ray Coherence formulation is designed to tackle this exactly; see lines 62-70 in the main paper. The robustness tests against different noise levels, shown in Table 2 of the main paper, can be regarded also as a test on how well our method handles potentially-mismatched keypoints/rays, since higher-level noise tends to introduce more mismatches.

Two reviewers requested experiments and comparisons on the LLFF dataset, as well as comparisons to more recent methods such as NoPe-NeRF, BAA-NGP, and NeuS2. To this end, we conducted these experiments and reported results in several new tables. The key conclusion is that our method, CRAYM, outperforms all the baselines and over all three metrics (PSNR, SSIM, and LPIPS). Please refer to the details in the individual rebuttals.

Other requested experiments can either be found in the supplementary material (e.g., on camera pose regression) or reported in the individual rebuttal (e.g., on sparse views). Again, our method comes on top.

All in all, we hope that the rebuttal will alleviate the concerns that the reviewers raised. We are happy to include as many new experimental results as space would allow in the revision. Code and any data used will be released upon paper publication.

-- The authors.

---

> ### Author Response · Authors · 2024-08-11
>
> Thank you to all the reviewers for your detailed and constructive feedback on our submission. We have addressed all the questions raised and conducted the required experiments. These primarily involved comparative analyses on the LLFF dataset, and the comparisons with BAA-NGP and NeuS2.
>
> In our experiments, the proposed method demonstrated superior performance across various metrics compared to the alternatives. The uploaded PDF file includes figures illustrating the optimized trajectories of the different methods evaluated.
>
> We remain open to addressing any further questions or concerns.

---

### Decision · Program_Chairs · 2024-09-25

**Decision:**

Accept (poster)

**Comment:**

This paper proposes a new method to jointly optimize camera poses and neural fields from multi-view images. The method incorporates contextual information and enforces geometric and photometric constraints through camera ray matching. The paper received one strong accept and three borderline accepts after the rebuttal, with most of the concerns raised by reviewers have been addressed.

Most reviewers acknowledged that the joint optimization along matching rays within the feature volume, as well as the design of key rays and auxiliary rays, effectively enhances volume optimization even in the presence of errors in matching and camera poses. Therefore, the paper is recommended for acceptance.